# Aggregation of ODC(I) and POL Defects in Bismuth Doped Silica Fiber

**DOI:** 10.3390/mi14020358

**Published:** 2023-01-31

**Authors:** Xiaofei Li, Binbin Wang, Tingting Song, Min Zhang, Tixian Zeng, Jiang Chen, Feiquan Zhang

**Affiliations:** 1School of Physics and Astronomy, China West Normal University, Nanchong 637002, China; 2College of Optoelectronic Engineering, Chengdu University of Information Technology, Chengdu 610225, China; 3Sichuan Hetai Optical Fiber Co., Ltd., Nanchong 637002, China

**Keywords:** bismuth doping, first principal calculation, ODC(I) defect, POL defect

## Abstract

First-principles calculations were used to simulate the aggregation of the peroxy chain defect POL and the oxygen vacancy defect ODC(I). Defect aggregation’s electronic structure and optical properties were investigated. The two defects were most likely to accumulate on a 6-membered ring in ortho-position. When the two defects are aggregated, it is discovered that 0.75 ev absorption peaks appear in the near-infrared band, which may be brought on by the addition of oxygen vacancy defect ODC(I). We can draw the conclusion that the absorption peak of the aggregation defect of ODC(I) defect and POL is more prominent in the near infrared region and visible light area than ODC(I) defect and POL defect.

## 1. Introduction

When pumped at 810 nm, the optical amplification of bi-doped silica glass at 1300 nm is close to the significant telecommunication window, and a novel broadband near-infrared (NIR) emission of bi-doped silica glass with long fluorescence lifespan has been described [1,2]. Success has been achieved in the demonstration of the continuous wave (CW) laser in bismuth-doped silica fiber, which can be applied to bi-doped fiber and associated devices [3]. Rare earth doped fibers, which are widely utilized in optical communication, medicine, material processing, and other industries, are also the most efficient active material as optical amplifiers. However, these rare earth-doped fiber lasers have one drawback: they cannot adequately cover the near infrared spectrum between 1150 and 1500 nm [4,5,6,7,8]. Thus, typical fiber amplifiers and lasers doped with rare earth ions do not meet the requirements of optical fiber communication. Bismuth-doped optical fibers and glasses have recently been discovered to have a distinctive broad-band IR luminescence in the range of 1000–1700 nm, with a full width at half maximum (FWHM) of about 300 nm and a service time of roughly hundreds of microseconds. These materials are successfully used in fiber lasing and amplification. The aforementioned disadvantage of rare-earth doped fiber amplifiers and lasers is expected to be compensated by bi-doped near-infrared (NIR) luminescent glass, which is also likely to usher in a new generation of materials for ultra-wideband optical amplification and laser media [9,10,11]. However, bismuth-doped optical fibers also have disadvantages, and the properties of Bi-related luminescence centers are not clear. This is mainly because the central valence state of bismuth atom in bismuth doped fiber is very variable. Because of the existence of oxidation-reduction equilibrium in the melting process and the reduction reaction in the high-temperature process. It is difficult to control this process with existing technology, which makes the bismuth atom in the bismuth doped fiber have multiple valence states. There are many hypotheses for the luminescence centers of bismuth-doped silica, such as: bismuth cluster ion: BiO [12], ${\mathrm{Bi}}_{2}^{-}$, ${\mathrm{Bi}}_{2}^{2-}$ [13,14], ${\mathrm{Bi}}_{2}^{4+}$, ${\mathrm{Bi}}_{5}^{3+}$ [15]; high-valence bismuth center: Bi^5+^ [16], Bi^3+^ [17,18]; low-valence bismuth center: Bi^2+^ [19], Bi^+^ [20,21,22,23,24] Bi^0^ [9]. In addition, it is simple to create a range of flaws during the manufacturing of silica glass optical fiber, each of which has a unique geometric structure, electronic properties, and optical properties. It is very challenging to study the luminescence properties of bismuth-doped silica fiber because the gap bismuth atoms are likely to encounter these defects during the doping process, causing significant changes in the local structure of these defects and seriously affecting the electronic properties and luminescence characteristics of bismuth active centers [25,26]. The two Frenkel defects, oxygen vacancy defect ODC(I) and peroxy chain linkage defect POL, are among the most significant flaws. The optical absorption of POL defects and ODC(I) defects are similar [27], and the reaction products of POL defects and bismuth atoms are more stable [12]. The combination produced by interstitial Bi^0^ and the defect ODC(I) is discovered to be able to produce near-infrared luminescence in the theoretical investigation of bismuth-doped silica glass [28]. The reaction pathway, action mechanism, optical properties, and electrical properties of defects in silica fiber have all been extensively studied. silica fiber’s performance, though, remains unclear when defects aggregate [29,30].

In this research, the first principle technique is used to calculate the formation energy, electrical characteristics, and optical properties in order to study the aggregation of oxygen vacancy defect ODC(I) and peroxy chain linkage defect POL in silica fiber. The fact that photons are significantly absorbed in the near-infrared range of 1000–1700 nm as a result of the aggregation of ODC(I) and POL in silica fiber has specific guiding relevance for the research and preparation of bismuth-doped silica fiber.

## 2. Model and Calculation Method

The amorphous silica fiber model was created using the traditional molecular dynamics (MD) method. The 2 × 2 × 2 crystal silica is simulated quenched to produce the amorphous silica glass model. When simulating the quenching process, we utilize the “NPT” (constant-pressure, constant-temperature) ensemble to regulate the temperature and set the force convergence precision to 0.01 eV/nm, LANGEVIN_GAMMA_L = 1. Tersoff potential was used to describe the interactions between atoms and the Langevin thermostat to control the system temperature. There are three steps in the melting quenching procedure. The heating stage that can guarantee the complete melting of the silica glass model is to first increase the temperature from 300 K to 6000 K in 0.01 ns. The system maintains a 0.01 ns transient equilibrium state at 6000 K in the second stage. In the final stage, the cooling mechanism lowers the temperature to 300 K in 0.01 ns. We set a time step of 1 fs for this process, 10,000 steps per procedure [19,20]. Amorphous silica has an average density of 2.39 g/cm^3^, which is in line with the findings of the experiments. 64 O atoms and 32 Si atoms make up the model [31,32].

The amorphous structure was then obtained, and its structure was optimized to construct defects and doped bismuth atoms based on the aforementioned defect-free amorphous silica model. The silicon atoms’ coordination number was then adjusted. We create defects by manually changing the position of silicon atoms and oxygen atoms, add defects and optimize the structure, then manually add a single bismuth atom and optimize the structure, Additionally, Frenkel defect ODC(I) and POL defect models were constructed, the aggregated defects are formed by adding POL to the silica glass model containing ODC(I) defects. To research the effects of combining two defect structures, the two defects were combined. In the context of density functional theory, the Vienna Ab initio Simulation Package (VASP) was used to carry out the first-principles calculations for this work (DFT). The electronic exchange and correlation interaction are described by the generalized gradient approximation (GGA) with the Perdew -Burke- Ernzerhof functional (PBE). The plane wave cut-off energy is set to 450 eV, the total energy is set to 10^−5^ eV, and the K-point is set to 2 × 2 × 2 [33].

## 3. Calculation Results and Discussion

### 3.1. Calculation of Doping Position

The structure of the amorphous silica glass optical fiber network is composed of a large number of tetrahedral units, including irregular element rings of different sizes. There are a lot of hybrid 3-element rings, hybrid 4-element rings, hybrid 6-element rings, and hybrid ring structural units for amorphous silica glass optical fiber [34]. The model’s doping formation energy was estimated to examine how the size of the ring structure affects the doping position. The following is a definition of the formation energy formula:(1)$$E_{f}(\mathrm{int}-\mathrm{Bi})=E_{\mathrm{tot}}(\mathrm{int}-\mathrm{Bi})-E_{tot}({\mathrm{SiO}}_{2},\mathrm{bulk})-u_{Bi}$$

The following symbols stand for the energies of doping formation, the energy of a defect-free silica glass model doped with a bismuth atom, and the energy of a defect-free silica glass model, respectively: *E_f_*(int-Bi), *E_tot_*(int-Bi), and *E_tot_*(SiO_2_,bulk) [35]. The chemical potential of the Bi atom, denoted by the symbol *u_Bi_*, the calculated value is −3.87 eV [36]. Bismuth atoms are doped in the gaps of three-membered rings, four-membered rings, five-membered rings and six-membered rings of the defect-free amorphous silica fiber, and their energies are calculated after the structure optimization. Similarly, the energy *E_tot_*(SiO_2_,bulk) of a defect-free silica glass model is calculated after structural optimization. The formation energy can be obtained by bringing the obtained data into the formation energy formula. The results are displayed in Table 1.

The defective energy formula is used to compute and count the As we can see, bismuth atoms are more likely to form six-membered rings since the model formation energy decreases with increasing ring size. The next calculation is the formation energy of defective amorphous silica model without bismuth atoms. A single ODC(I) defect model is used as the initial model to add POL defects at various locations in order to examine the properties of POL defects and ODC(I) aggregation in various sizes of element rings. The 96-atom silica fiber model is depicted in Figure 1a,b with the defects ODC(I)and POL, respectively. In which red represents oxygen atoms and blue represents silicon atoms. Two defects in the adjacent position are called the ortho-position. When two defects are in the symmetrical position of the ring, they are called the para-position. When the defects are separated by an oxygen atom, they are called the meta-position. There are two different forms of defects in a 4-membered ring, known as 4-mr-para and 4-mr-ortho, as seen in Figure 1c,d, respectively. As shown in Figure 1e,f, the five-membered ring has two ortho- and meta-position situations, respectively known as 5mr-meta and 5mr-ortho. As demonstrated in Figure 1g–i, which are referred to as 6mr-para, 6mr-meta, and 6mr-ortho, the defective positions in the six-membered ring might be in the opposing, intermediate, and neighboring positions.

The formation can be used to illustrate how changing it is to combine two model flaws. The formation of the defect can be described by the following formula when a POL defect and an ODC(I) defect are present in the model at the same time:(2)$$E_{f}=E_{ODC\left(I\right)POL}-E_{\mathrm{perfect}}$$
where *E_f_* stands for the energy that leads to the production of defects, *E_ODC(I)POL_* for the energy that includes both ODC(I) defects and POL defects, and *E_perfect_* for the energy of the defect-free model [37]. The defect formation energy of two types of defects in various places, from a 4-membered ring to a 6-membered ring, is computed using the defect formation energy formula, and the formation energy. In order to study the relationship between bond length variation and formation energy, we need to calculate the bond lengths of silicon-silicon and oxygen-oxygen bonds [38], as shown in Table 2. According to the references, In the construction of ODC(I) and POL defects, the silicon-silicon bond length of ODC(I) and the oxygen-oxygen bond of POL defects are set to 2.44 Å [27] for ODC(I) and 1.38 ± 0.1 Å [39] for POL defects, respectively. The table shows that the two defects are more likely to appear in the ortho-position for the 4-membered ring and the 6-membered ring whereas they are more likely to appear in the meta-position for the two defects in the 5-membered ring. The two defects created in the ortho-position and meta-position of the five-membered ring and the ortho-position of the six-membered ring have lower formation energies, suggesting that the two types of defect aggregation are more likely to occur in these two configurations. The lengths of the silicon-silicon bond and the oxygen-oxygen link are not linearly related to the formation energy, which means there is no clear connection between the formation energy and these bond lengths.

### 3.2. Analysis of Electronic and Optical Properties

The electronic properties and optical properties of the six-membered ring were computed using the aggregation model due to the low formation energy of the two defects in the ortho-position. The density of states and optical properties in the case of doping solely bismuth atoms in fiber were determined in order to examine the impact of bismuth atoms and aggregation defects on fiber. The density of states of intrinsic defect-free silica fibers is shown in Figure 2a. In this paper, the band gap of the defect-free silica fiber model is set at 4.961 eV, which is consistent with the band gap calculated in the literature as 5 eV by the GGA-PBE method [40]. The red line shows the total electronic density of states in silica fiber doped with bismuth, while the black line shows the total electronic density of states in defect-free silica fiber. There are three obvious unoccupied defect states in the total electronic density of states of gap bismuth-doped silica fiber, which are located at 1.4 eV, 3.9 eV, and 4.8 eV, respectively. And this is discovered by comparing the total electronic density of states of defect-free silica fiber and the total electronic density of states of bismuth-doped silica fiber. The partial density of states of gap bismuth-doped silica fiber is calculated in order to further analyze how atomic orbitals affect unoccupied defect states, as illustrated in Figure 2b. The 6p orbital of the bismuth atom is shown by the red line, the 2p orbital of oxygen is represented by the green line, and the 3p orbital of silicon is represented by the blue line. According to the electron total density of states diagram for the bismuth-doped silica fiber, the silicon atom’s 3p orbital contributes very little to the three unoccupied defect states at 1.4 eV, 3.9 eV, and 4.8 eV. The main contributors to these states are the bismuth atom’s 6p orbital and the partial contributors is the oxygen atom’s 2p orbital. The appearance of these unoccupied defect states leads to the reduction in the model band gap of bismuth-doped silica fiber, which is conducive to the photon absorption of bismuth doped-silica fiber. As a result, the optical characteristics of bismuth-doped silica fiber are primarily determined by the 6p orbital of the introduced gap bismuth atom and may be slightly influenced by the adjacent oxygen atom’s 2p orbital.

In this section, we primarily look into how defects in bi-doped silica optical fiber relate to their electronic properties. First, we calculate the Bader charge. When the bismuth atom is not doped, the effective charge of the silicon atom on the silicon-silicon bond is +2.41 |e| and +2.36 |e|, and the effective charge of the oxygen atom on the oxygen-oxygen bond is −0.86 |e| and −0.76 |e|. After doping bismuth, the effective charge of the silicon atom on the silicon-silicon bond is +2.37 |e| and +2.47 |e|, and the effective charge of the oxygen atom on the oxygen-oxygen bond is −1.1 |e| and −1.36 |e|. The comparison shows that the charge of the silicon atom is almost unchanged, while the bismuth atom transfers −0.93 |e| charge to the oxygen atom. At this time, the bismuth atom is almost univalent. The density of states for the ODC(I) defect, POL defect, and these two defects in the ortho-position of the six-membered ring are shown in Figure 3a–c for both silica glass without defects and silica glass that has been doped with bismuth. The unoccupied defect states for the bismuth-doped silica glass model with the ODC(I) defect are 0.55 eV and 2.45 eV in Figure 3a. According to Figure 3b [41,42,43], for the bismuth-doped silica glass model with POL defect, the unoccupied defect states appear at 1.14 eV and 3.0 eV. When the POL defect and ODC(I) defect aggregate in the bi-doped silica optical fiber, the unoccupied defect states can be seen in Figure 3c at 0.2 eV and 0.42 eV. Unoccupied defect states are also seen at 0.2 eV and 3.95 eV when POL defects and ODC(I) defects are aggregated in defect-free silica fibers. The calculated DOS presented in Figure 3, suggests that the enhanced optical absorption at 1600–1700 nm in bi-doped silica based fibers can possibly be determined by the monovalent Bi atom aggregated with ODC(I) and POL intrinsic defects.

Figure 4a depicts a hypothetical part diagram for the silica fiber’s permittivity, and Figure 4b depicts the silica fiber’s absorption spectrum. First, the structure of the model is optimized, then the static self-consistent calculation is performed, and finally the optical properties are calculated. The smooth curve of a silica fiber free of flaws is devoid of an absorption peak. The Figure 4 shows that the bi-doped silica fiber, the bismuth-doped ODC(I) defect model, the bismuth-doped POL defect model, and the aggregated bismuth-doped silica fiber model with two defects have absorption peaks in the visible and near-infrared regions [44]. In both defect aggregation cases, the absorption peaks of the bismuth-doped silica fiber model are significantly higher than those of other fibers in the visible and near-infrared regions. In the case of defect aggregation, the absorption peak reached 1.13 at 0.75 eV, while the absorption peak of POL defect was only 0.08 at 1.07 eV, this is because the bismuth atom is monovalent after the reaction between the bismuth atom and the aggregation defect, resulting in enhanced optical absorption at 1600–1700 nm.

## 4. Conclusions

In this paper, first-principles calculations are used to investigate the oxygen vacancy defects ODC(I), peroxy chain defects POL, and their aggregation properties in bismuth-doped silica fibers. Calculations are made regarding the formation of energies, electronic structures, and absorption spectra. The formation energy of bismuth atoms doped in different rings was calculated, and the results indicate that the formation energy of bismuth atoms doped in larger rings is lower. We calculate the formation energies of two different types of defect aggregation at various locations in four- to six-membered rings. As a result, in the ortho-position of six-membered rings, the formation energy of two types of defect aggregation is lower. The silica fiber with bismuth doping alone is then calculated, and its results are compared to those of the defect-free silica fiber. The bismuth-doped silica fiber was discovered to have three unoccupied defect states at 1.4 eV, 3.9 eV, and 4.8 eV. Finally, the electronic and optical properties of a single defect and two kinds of defect aggregation were calculated in the ortho-position of a six-membered ring of silica fiber. In the presence of the six-membered ring of silica fiber in ortho-position, it is discovered that there are obvious absorption peaks in the visible and near-infrared regions. The absorption peaks at 0.75 eV are in the near-infrared band of 1600–1700 nm. Our calculations have significant theoretical guiding implications for the thorough examination of the interaction mechanism between defects and bismuth-doped silica fiber as well as for the research and preparation process of bismuth-doped fiber.

## Figures and Tables

**Figure 1 micromachines-14-00358-f001:**
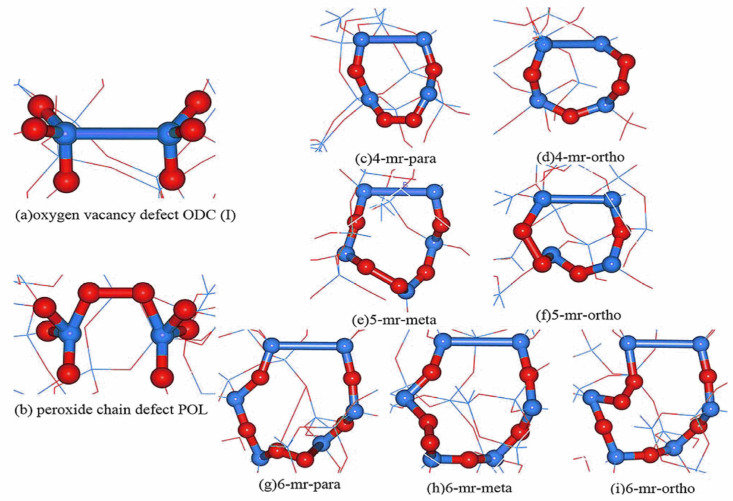
(**a**) oxygen vacancy defect ODC(I) Structure diagram (**b**) peroxide chain defect POL Structure diagram. (**c**–**i**) are the Co-doping model of peroxy bond defect (POL) and oxygen vacancy defect (ODC(I)), (**c**,**d**) are the para-position and ortho-position models of two defects in a 4-membered; (**e**,**f**) are the meta-position and ortho-position models of two defects in a 5-membered ring: (**g**–**i**) are the para-position, meta-position and ortho-position models of two defects in a 6-membered ring. Blue ball is silicon atom, red ball is oxygen atom. Blue ball is silicon atom, red ball is oxygen atom.

**Figure 2 micromachines-14-00358-f002:**
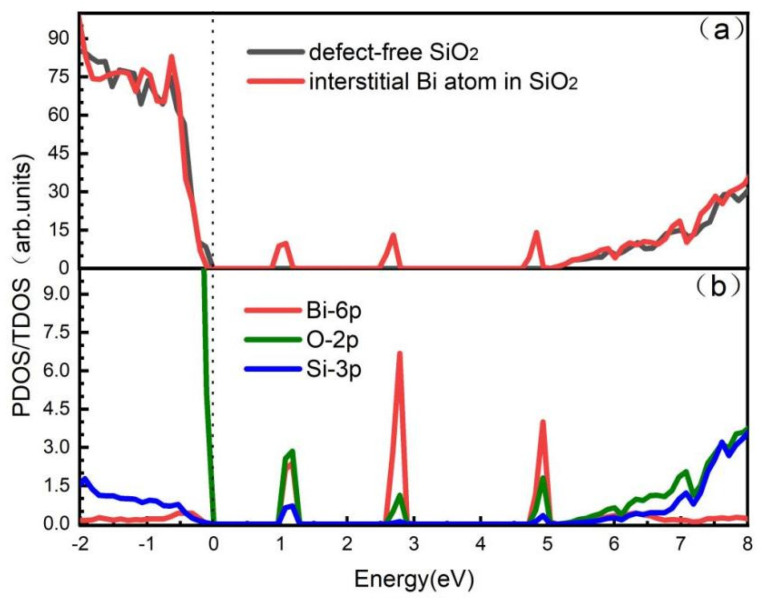
(**a**) Black represents the total electronic density of states of defect-free silica fiber, and red represents the total electronic density of states of bismuth-doped silica fiber. (**b**) Red represents the 6p orbital of bismuth atom, green represents the 2p orbital of oxygen atom, and blue represents the 3p orbital of silicon atom.

**Figure 3 micromachines-14-00358-f003:**
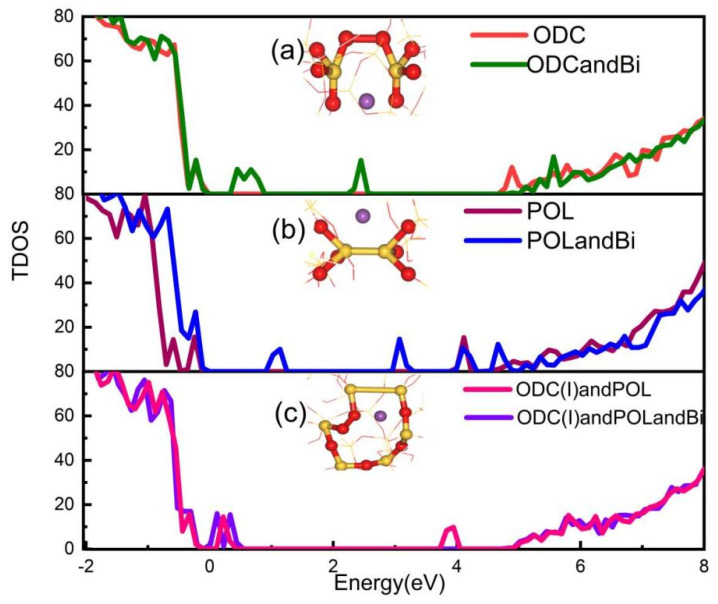
(**a**) Red line represents the density of states diagram of Oxygen vacancy defect ODC(I) of Undoped silica fiber, and blue line represents the density of states diagram of bismuth-doped silica fiber of Oxygen vacancy defect ODC(I). (**b**)wine red line represents the density of states diagram of peroxy chain defect POL of Undoped silica fiber, and blue line represents the density of states diagram of peroxy chain defect POL of bismuth-doped silica fiber. (**c**) pink line represents the density of states diagram of undoped silica fiber with oxygen vacancy defect ODC(I) and peroxy bond defect (POL) presence in the ortho-position of the six-membered ring, and purple line represents the density of states diagram of bismuth-doped silica fiber with oxygen vacancy defect ODC(I) and peroxy bond defect (POL) presence in the ortho-position of the six-membered ring.

**Figure 4 micromachines-14-00358-f004:**
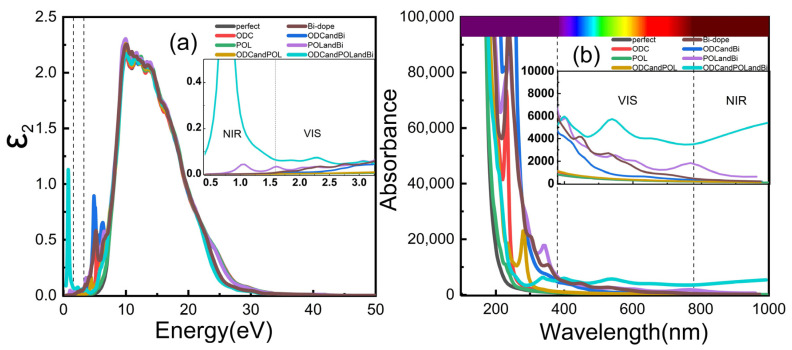
(**a**) Diagram of the imaginary part of the permittivity of silica fiber. (**b**) Absorption spectrum of silica fiber.

**Table 1 micromachines-14-00358-t001:** Distribution table of formation energy of bismuth atom gap doping.

n-Membered Rings	3-MR	4-MR	5-MR	6-MR
Formation Energy (eV)	9.16	14.32	4.5	3.81

**Table 2 micromachines-14-00358-t002:** Formation energy and structure of ODC(I) defects and POL defects added in different ring positions and bond length of silicon-silicon bond and oxygen-oxygen bond after structural optimization.

Meta Ring	Model	Forming Energy (eV)	Si-Si (Å)	O-O (Å)
4MR	Para-position Ortho-position	7.13 7.02	2.424 2.481	1.514 1.520
5MR	Meta-position Ortho-position	6.84 6.94	2.430 2.412	1.474 1.479
6MR	Para-position Ortho-position Meta-position	7.91 6.43 7.85	2.561 2.464 2.534	1.476 1.510 1.490

## Data Availability

Not applicable.

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
