# Peer review of "Aggregation of ODC(I) and POL Defects in Bismuth Doped Silica Fiber"

_micromachines, 2023, doi:10.3390/mi14020358_

Round 1
Author Response
- We hope the authors briefly clarify the rationale for selecting the peroxy chain connection defects and oxygen vacancy defects as the research objects, as well as the distinctions between these two defects and other defects.
Answer:The optical absorption of POL defects and ODC(I) defects are similar, and the reaction products of POL defects and bismuth atoms are more stable[1,2].
[1] Imai H, Arai K, Imagawa H, et al. Two types of oxygen-deficient centers in synthetic silica glass[J]. Physical Review B, 1988, 38(17): 12772.
[2] Han L, Xiao W, Zhang J, et al. Investigations of interstitial Bi0 interacting with intrinsic defects in bismuth-doped silica optical fiber[J]. Applied Physics A, 2019, 125(9): 1-7.
- The model's doping formation energy was estimated to examine how the size of the ring structure affects the doping position. While there is no clear connection between the formation energy and these bond lengths. Why does the authors focus on the study of bond length?
Answer:in order to study the relationship between bond length variation and formation energy, we need to calculate the bond lengths of silicon-silicon and oxygen-oxygen bonds[1].
[1]Jia B, Guan Z, Peng Z, et al. Structural disorder in fused silica with ODC (I) defect[J]. Applied Physics A, 2018, 124(10): 1-6.
- We recommend that the authors combine the Bader charge density to undertake a full research and description of the charge transfer in the Bi ions doping process.
Answer:Thanks a lot for your kind suggestion, We have calculated the Bader charge and added it to the article.
- If feasible, please describe the microscopic mechanism by which the absorption peak of Bi ions doped quartz fiber in the infrared and visible light bands is much higher than that of other fibers.
Answer:this is because the bismuth atom is monovalent after the reaction between the bismuth atom and the aggregation defect
- The absorption peaks of different optical fibers in the infrared and visible light regions are presented for comparison to demonstrate the conclusion's validity.
Answer:Thanks a lot for your kind suggestion,we have added relevant content in the article.

Reviewer 2 Report
The authors reported the aggregation properties of oxygen vacancy defect ODC (I) and peroxy chain defect POL in bismuth-doped silica fibers in the framework of DFT. They investigated the formation energy, the electronic and optical properties of defects. This manuscript was generally well-organized. Nevertheless, there are some problems observed in this manuscript, which should be carefully addressed. For these reasons, I can only agree to a publication of this manuscript in Micromachines after a major revision.

Author Response
- In section 3.1, the authors calculated the formation energy of 3MR, 4MR, 5MR, and 6MR. They found that the 6MR is more likely to be formed. Did they calculate the formation energy of larger nMR, such as 7MR or 8MR?
Answer:3MR, 4MR, 5MR, and 6MR-rings are more prevalent in silica fibers
The atomic models of Bi-doped and ODC(I)/POL/Bi-co-doped quartz fiber should be present.
Answer: We can get the aggregation defect may exist by calculation formation and the bismuth atom reacts with it in the valence state of monovalent.
- In section 3.1, the meaning of para, meta, and ortho should be specified, when they occurred for the first time.
Answer:Thanks a lot for your kind suggestion,we have added relevant content in the article.
- It is recommended to add dotted lines in Table 2 to make it clearer.
Answer:Thanks a lot for your kind suggestion,We have modified the table.
- In the last paragraph in Section 3.2, the authors should explain this part more deeply.
Answer:Thanks a lot for your kind suggestion,we have added relevant content in the article.
- 6. In Conclusion, the absorption peak at 0.75 eV is not in the near-infrared band of 1400-1500 nm.
Answer:This error has been modified.
- How did the author treat the spin-orbital coupling effect in the calculation?
Answer:Magnetism has little influence on the system I study, so I did not consider the influence of orbital spin coupling on the system.
- The manuscript is generally well-written. However, some errors
still can be found:
On line 36, “by Bi-doped near-infrared (NIR) luminescent glass”.
On line 78, “10-5 eV”.
On line 144, “total total”.
On line 153, “3p orbital”.
Answer:These errors have been modified.

Reviewer 3 Report
The work is interesting, but the article is stupidly written.
The authors should make some corrections in the text before the manuscript will be considered for the publication again.
I do not offer any additional computer simulation, but ask the authors only to describe the methods used and the results obtained more clearly.
Comments.
1. Introduction:
1. The authors used the melting quenching method to obtained a model of amorphous (glassy) silica.
The should read two recent papers devoted to the same problem and give references on them:
V.B. Sulimov, D.C. Kutov, A.V. Sulimov, F.V. Grigoriev, A.V. Tikhonravov,
Density functional modeling of structural and electronic properties of amorphous high temperature oxides,
Journal of Non-Crystalline Solids,
Volume 578,
2022,
121170,
ISSN 0022-3093,
https://doi.org/10.1016/j.jnoncrysol.2021.121170.
Sulimov, A.V., Kutov, D.C., Grigoriev, F.V. et al. Generation of Amorphous Silicon Dioxide Structures via Melting-Quenching Density Functional Modeling. Lobachevskii J Math 41, 1581–1590 (2020).
https://doi.org/10.1134/S1995080220080193
2. There is a misprint in citation in Introduction: after ref [1,2] there is [7]! Refs 3,4,5, and 6 which are in the list of references
are not cited in the text!
3. It is better to use "silica fiber" instead of "quartz fiber"
4. In Introduction, authors should discuss in details existing publications on Bi states in glasses. In particular to discuss ref. [14], where the aggregation of Bi with oxygen vavancy was concluded to be the most probable Bi-related IR luminescence center.
5. Authors wrote: "The combination produced by interstitial Bi0 and the defect ODC (I) is discovered to be able to produce near-infrared luminescence in the theoretical investigation of bismuth-doped silicon glass [16]." However, there is nothing about bismuth-doped silica glass in ref. [16}. Also. there is no "...theoretical investigation..." in [16].
6. Authors should include in Introduction additional references related to models of Bi IR luminescence centre:
6.1 Alexey N. Romanov, Zukhra T. Fattakhova, Denis M. Zhigunov, Vladimir N. Korchak, Vladimir B. Sulimov,
"On the origin of near-IR luminescence in Bi-doped materials (I). Generation
of low-valence bismuth species by Bi3+ and Bi0 synproportionation", Optical Materials 33 (2011) 631–634.
6.2 Alexey N. Romanov, Zukhra T. Fattakhova, Alexander A. Veber, Olga V. Usovich,
Elena V. Haula, Vladimir N. Korchak, Vladimir B. Tsvetkov, Lev A. Trusov, Pavel E. Kazin, and Vladimir B. Sulimov
"On the origin of near-IR luminescence in Bi-doped materials (II). Subvalent monocation
Bi+ and cluster Bi_5^3+ luminescence in AlCl3/ZnCl2/BiCl3 chloride glass",
OPTICS EXPRESS (2012) Vol. 20, No. 7, 7212.
6.3 Alexey N. Romanov, Elena V. Haula, Zukhra T. Fattakhova, Alexander A. Veber,
Vladimir B. Tsvetkov, Denis M. Zhigunov, Vladimir N. Korchak, Vladimir B. Sulimov
"Near-IR luminescence from subvalent bismuth species in fluoride glass", Optical Materials 34 (2011) 155–158.
6.4 A. A. Veber, A. N. Romanov, O. V. Usovich, Z. T. Fattakhova,
E. V. Haula, V. N. Korchak, L. A. Trusov, P. E. Kazin,
V. B. Sulimov, V. B. Tsvetkov "Luminescent properties of Bi-doped polycrystalline KAlCl4", Appl. Phys. B (2012) 108:733–736.
6.5 A. N. Romanov, A. A. Veber, D. N. Vtyurina, Z. T. Fattakhova, E. V. Haula,
D. P. Shashkin, V. B. Sulimov, V. B. Tsvetkov and V. N. Korchak "Near infrared photoluminescence of the univalent
bismuth impurity center in leucite and pollucite crystal hosts", Journal of Materials Chemistry C (2015), 3, 3592.
6.6 A. N. Romanov et al., "New route to Bi+-doped crystals: Preparation and NIR luminescence of K, Rb and Cs ternary chlorides, containing univalent bismuth," 2013 Conference on Lasers & Electro-Optics Europe & International Quantum Electronics Conference CLEO EUROPE/IQEC, Munich, Germany, 2013, pp. 1-1, doi: 10.1109/CLEOE-IQEC.2013.6800996.
6.7 D. N. Vtyurina, A. N. Romanov, K. S. Zaramenskikh, M. N. Vasil’eva, Z. T. Fattakhova,
L. A. Trusov, P. A. Loiko, and V. N. Korchak "IR Luminescence of Bismuth-Containing Centers in Materials
Prepared by Impregnation and Thermal Treatment of Porous Glasses", Russian Journal of Physical Chemistry B, 2016, Vol. 10, No. 2, pp. 211–214.
6.8 A.N.Romanov, E.V.Haula,D.P.Shashkin,D.N.Vtyurina,V.N.Korchak
"On the origin of near-IR luminescence in SiO2 glass with bismuth as the
single dopant.Formation of the photoluminescent univalent bismuth
silanolate by SiO2 surface modification. Journal ofLuminescence 183(2017)233–237.
Actually, all refs. [6.1 - 6.8] are efforts to prove the nature of Bi-related broad IR luminescence.
Almost the same features of Bi-related IR luminescence in different glasses and crystals support the models of bismuth monocations and
subvalent polycations. In any case, the discussion of these experimentally confirmed results in the Introduction is extremely important.
2. Model and calculation method:
1. Line 68: "Amorphous silica has an average density of 2.39g/cm3,..." How many steps are used to perform the averaging? Add this information to the text.
2. The density of 2.39 g/cm3 is too high comparing with silica glass (2.20 g/cm3). Why?
3. Lines 71 - 74: authors must write more clearly how defects were constructed in the obtained amorphous silica. Were intrinsic defects created by hand or they were obtained in the melting-quenching process? How Bi atoms were introduced into the glass matrix? Were Bi atoms introduced in the quartz crystal before melting-quenching or after?
"Additionally, Frenkel defect ODC (I) and POL defect models were constructed". Authors should describe how these defects were constructed.
4. What thermostat was used in the MD simulations? The time step 1 fs (instead of 1FS - line 68)!
5. Were the PBE functional and the projector-augmented wave (PAW) method used? Authors should present this information.
6. Where did the authors get the crystal structure of quartz from?
7. Authors should present values of all LANGEVIN_GAMMA parameters, if Langevin thermostat was used. Or present another parameters, if another thermostat was used.
3.1. Calculation of doping position
1. Lines 89 - 90: "The chemical potential of the Bi atom, denoted by the symbol uBi, is calculated to be -3.87eV [26]." There is nothing about Bi in [26]!!! Authors should explain where did they take value of uBi -3.87 eV from?
2. Authors should describe how did they created Bi-centers near rings! Were there MD simulations or energy optimization after introducing Bi atom? How Bi centers were created? Is Bi-center a Bi interstitial atom or a Bi cluster of several Bi atoms? How values in Table 1 were obtained: which values were used in eq. (1)?
3. How ODC and POL defects were created?
4.Lines 121 - 124: "According to the references, In the construction of ODC (I) and POL defects, the silicon-silicon bond length of ODC (I) and the oxygen-oxygen bond of POL defects are set to 2.44 Å [28] for ODC (I) and 1.38±0.1 Å [29] for POL defects, respectively.". Does it mean that ODC(I) and POL are constructed by hand? They did not obtained by melting-quenching. Were Si-Si and O-O distances presented in Table 2 obtained after MD simulations or energy optimization? What type of energy optimization was used?
3.2. Analysis of electronic and optical properties
1. There results on values of energy excitations calculated by PBE are much low than experimental obsevations. The energy gap of defect free amorphous state of silica obtained by authors (4.961 eV) is much lower than experimentally observed value of 9.2 eV for silicon dioxide. Authors wrote "The ab-216 sorption peaks at 0.75ev are in the near-infrared band of 1400–1500nm." Usually, positions of peaks of DOS in the energy gap do not directly connected to absorption or luminescence bands in PBE calculations! Authors shour correct the corresponding text.
2. Line 176 -179: "When the POL defect and ODC (I) defect aggregate in the bi-doped silica optical fiber, the unoccupied defect states can be seen in Figure 3(c) at 0.2eV and 0.42eV. Unoccupied defect states are also seen at 0.2 eV and 3.95 eV when POL defects and ODC(I) defects are aggregated in defect-free silica fibers."
The authors should explain and present a figure with the structure of an aggregate of POL and ODC(I) defects with Bi and without Bi.
Author Response
Review 3
- Introduction:
1.The authors used the melting quenching method to obtained a model of amorphous (glassy) silica.The should read two recent papers devoted to the same problem and give references on them:
V.B. Sulimov, D.C. Kutov, A.V. Sulimov, F.V. Grigoriev, A.V. Tikhonravov,Density functional modeling of structural and electronic properties of amorphous high temperature oxides,Journal of Non-CrystallineSolids,Volume 578,2022,121170,ISSN 0022-3093, https://doi.org/10.1016/j.jnoncrysol.2021.121170.
Sulimov, A.V., Kutov, D.C., Grigoriev, F.V. et al. Generation of Amorphous Silicon Dioxide Structures via Melting-Quenching Density Functional Modeling. Lobachevskii J Math 41, 1581–1590 (2020).https://doi.org/10.1134/S1995080220080193
Answer:Thank you for your recommendation. We have read these two articles and added them to the references.
- There is a misprint in citation in Introduction: after ref [1,2] there is [7]! Refs 3,4,5, and 6 which are in the list of references
are not cited in the text!
Answer:This error has been modified.
- It is better to use "silica fiber" instead of "quartz fiber"
Answer:This error has been modified.
- In Introduction, authors should discuss in details existing publications on Bi states in glasses. In particular to discuss ref. [14], where the aggregation of Bi with oxygen vavancy was concluded to be the most probable Bi-related IR luminescence center.
Answer:Thank you for your suggestion, Bismuth-doped fiber related articles have been discussed and added to the article.
- Authors wrote: "The combination produced by interstitial Bi0 and the defect ODC (I) is discovered to be able to produce near-infrared luminescence in the theoretical investigation of bismuth-doped silicon glass [16]." However, there is nothing about bismuth-doped silica glass in ref. [16}. Also. there is no "...theoretical investigation..." in [16].
Answer:This error has been modified.
- Authors should include in Introduction additional references related to models of Bi IR luminescence centre:
Actually, all refs. [6.1 - 6.8] are efforts to prove the nature of Bi-related broad IR luminescence.Almost the same features of Bi-related IR luminescence in different glasses and crystals support the models of bismuth monocations andsubvalent polycations. In any case, the discussion of these experimentally confirmed results in the Introduction is extremely important.
Answer:Thank you for your recommendation. We have read these two documents and added them to the references.
- Model and calculation method:
- Line 68: "Amorphous silica has an average density of 2.39g/cm3,..." How many steps are used to perform the averaging? Add this information to the text.
Answer:Thanks a lot for your kind suggestion, we have added relevant content in the article.
- The density of 2.39 g/cm3 is too high comparing with silica glass (2.20 g/cm3). Why?
Answer:This is because we use Stishovite to construct silica glass, while the reference paper uses alpha silica. The density difference is within a reasonable range.
- Lines 71 - 74: authors must write more clearly how defects were constructed in the obtained amorphous silica. Were intrinsic defects created by hand or they were obtained in the melting-quenching process? How Bi atoms were introduced into the glass matrix? Were Bi atoms introduced in the quartz crystal before melting-quenching or after?
"Additionally, Frenkel defect ODC (I) and POL defect models were constructed". Authors should describe how these defects were constructed.
Answer:Thanks a lot for your kind suggestion, we have added relevant content in the article.
“The amorphous structure was then obtained, and its structure was optimized to construct defects and doped bismuth atoms based on the aforementioned defect-free quartz model. The silicon atoms' coordination number was then adjusted. We create defects by manually changing the position of silicon atoms and oxygen atoms, add de-fects and optimize the structure, then manually add a single bismuth atom and opti-mize the structure, Additionally, Frenkel defect ODC (I) and POL defect models were constructed, the aggregated defects are formed by adding POL to the quartz model containing ODC (I) defects. To research the effects of combining two defect structures, the two defects were combined.”
- What thermostat was used in the MD simulations? The time step 1 fs (instead of 1FS - line 68)!
Answer:The Tersoff potential was used to describe the interactions between atoms and the Langevin thermostat to control the system temperature. This error has been modified.
- Were the PBE functional and the projector-augmented wave (PAW) method used? Authors should present this information.
Answer:The electronic exchange and correlation interaction are described by the generalized gradient approximation (GGA) with the Perdew-Burke-Ernzerhof functional (PBE).
- Where did the authors get the crystal structure of quartz from?
Answer: The quartz structure is directly downloaded from the materials studio database.
- 7. Authors should present values of all LANGEVIN_GAMMA parameters, if Langevin thermostat was used. Or present another parameters, if another thermostat was used.
Answer:LANGEVIN_GAMMA_L=1,,we have added relevant content in the article.
3.1. Calculation of doping position
- Lines 89 - 90: "The chemical potential of the Bi atom, denoted by the symbol uBi, is calculated to be -3.87eV [26]." There is nothing about Bi in [26]!!! Authors should explain where did they take value of uBi -3.87 eV from?
Answer:This error has been modified.
- Authors should describe how did they created Bi-centers near rings! Were there MD simulations or energy optimization after introducing Bi atom? How Bi centers were created? Is Bi-center a Bi interstitial atom or a Bi cluster of several Bi atoms? How values in Table 1 were obtained: which values were used in eq. (1)?
Answer:Thanks a lot for your kind suggestion,we have added relevant content in the article.
- How ODC and POL defects were created?
Answer:Defects are formed by adjusting the position of atoms.
4.Lines 121 - 124: "According to the references, In the construction of ODC (I) and POL defects, the silicon-silicon bond length of ODC (I) and the oxygen-oxygen bond of POL defects are set to 2.44 Å [28] for ODC (I) and 1.38±0.1 Å [29] for POL defects, respectively.". Does it mean that ODC(I) and POL are constructed by hand? They did not obtained by melting-quenching. Were Si-Si and O-O distances presented in Table 2 obtained after MD simulations or energy optimization? What type of energy optimization was used?
Answer:The defect is obtained through manual construction and structural optimization. The measured atomic distance is also the measured atomic distance after structural optimization,we have added relevant content in the article.
3.2. Analysis of electronic and optical properties
- There results on values of energy excitations calculated by PBE are much low than experimental obsevations. The energy gap of defect free amorphous state of silica obtained by authors (4.961 eV) is much lower than experimentally observed value of 9.2 eV for silicon dioxide. Authors wrote "The ab-216 sorption peaks at 0.75ev are in the near-infrared band of 1400–1500nm." Usually, positions of peaks of DOS in the energy gap do not directly connected to absorption or luminescence bands in PBE calculations! Authors shour correct the corresponding text.
Answer:This error has been modified.
- Line 176 -179: "When the POL defect and ODC (I) defect aggregate in the bi-doped silica optical fiber, the unoccupied defect states can be seen in Figure 3(c) at 0.2eV and 0.42eV. Unoccupied defect states are also seen at 0.2 eV and 3.95 eV when POL defects and ODC(I) defects are aggregated in defect-free silica fibers."
The authors should explain and present a figure with the structure of an aggregate of POL and ODC(I) defects with Bi and without Bi.
Answer:Thanks a lot for your kind suggestion, we have added relevant content in the article. Please see fig3.

Round 2
Reviewer 1 Report
The authors have basically addressed my concerns. So I agree to publish the revisied manuscript in Micromachines.
Author Response
Thank you for your review and comments.Reviewer 2 Report
The authors included all my suggestions how to improve the manuscrit into the revised version of the paper . The paper now is suitable for publication . I recommend publication. The paper may be accepted as it is , without any further changes .
Author Response
Thank you for your review and comments.Reviewer 3 Report
Comments, 2nd round.
1. Line 23: instead of "silicon glass" you must write "silica glass". You must make this correction throughout the manuscript.
2. Line 26: instead of "...fiber, which is..." you must write "...fibers, which are..."
3. Line 27: instead of "...is..." you must write "...are also..."
4. Line 28: instead "...a reinforcer." you must write "...optical amplifiers."
5. Lines 28 - 30: instead of "These rare earth-doped fiber lasers do have one drawback, though, 28 which is that they are unable to adequately span the near-infrared spectrum between 1150 29 and 1500 nm" you must write "However, these rare earth-doped fiber lasers have one drawback: they cannot adequately cover the near infrared spectrum between 1150-29 and 1500 nm."
6.Lines 30, 31: instead of "Therefore, the typical rare earth ion doped fiber amplifier and laser 30 hence have trouble meeting the demands of optical fiber communication." you must write "Thus, typical fiber amplifiers and lasers doped with rare earth ions do not meet the requirements of optical fiber communication."
7. Lines 35 -38: instead of "The lack of rare earth ion doped fiber amplifiers and lasers is expected to be compensated for by Bi-doped near infrared (NIR) luminescent glass, which will also likely usher in a new generation of ultra wideband optical amplification materials and laser media [9-11]." you must write "The aforementioned disadvantage of rare-earth doped fiber amplifiers and lasers is expected to be compensated by Bi-doped near-infrared (NIR) luminescent glass, which is also likely to usher in a new generation of materials for ultra-wideband optical amplification and laser media [9-11]."
8. Line 46: BiO[Error! Reference source not found.] Please, correct this error.
9. Lines 46 - 62: There are some errors with numbers of references. The refs. 12, 13, 15, 19, 20, 21, 22, 23 are omitted.
Refs with numbers 12, 13, 15, 19, 20, 21, 22, 23 are missing.
10. Line 72: instead of "quartz" you must write "silica"
11. Line 80: instead of "1FS" you must write "1 fs"
12. Line 81: Add information about time (or number of steps) of stabilization the system at 300 K; Add the information on how many steps are used to perform the averaging to obtain density of 2.39 g/cm3.
13. Line 85: instead of "quartz" you must write "amorphous silica".
14. Line 90: instead of "quartz model" you must write "silica glass model". You must make this correction throughout the manuscript.
15. Line 96: you must change "ev" to "eV"
16. Line 103: correct the ref [306]: there is no such big number in the List of references!
17. Line 110: Correct English!
18. Line 126: "formation energy of three-, four-, five-, and six-membered rings. The results are displayed in Table 1." The English sentence is not full, and some words are missed!
19. Line 128: "Table 1. Distribution table of formation energy of bismuth atom gap doping."
"...bismuth atom gap doping": What does it mean?
20. Is Table 1 for Bi-free system or for Bi-doped system?
21. If values in Table 1 correspond to the system with Bi atom, write clearly in the text: where was this bismuth atom initially placed before energy optimization?
22. To write where Bi atom is in Figure 1!!!
23. It is still not clear how the values presented in Table 1 were obtained. You must write in the text clearly how values in Table 1 were obtained using equation (1).
24. Line 157: to write clearly are values presented in Table 2 related to Bi-free samples! What does it mean "...ODC(I) defects and POL defects doped in different ring positions..."? What does it mean here "...doped..."? Is the doping Bi?
25. You must explain how curves in Fig. 4 were obtained. Were they calculated or experimentally measured?
If they were calculated, you must write how the calculations were made.
26. Line 230: you must to write how you calculated the position of the absorption peak at 0.75 eV. If it was obtained but a simple difference of peaks in the Density Of States (DOS) in Fig.3, it is incorrect!!! In general, you cannot calculated exact position of the absorption band from peaks positions in DOS! The calculation of the Energy gap (the different between LUMO and HOMO states in defect free silica glass using PBE functional in about 5 eV instead of 9 eV !
27. You must to improve English throughout the manuscript!
28. In response to reviewer's comment, which I wrote in the previous time:
"Authors wrote "The absorption peaks at 0.75ev are in the near-infrared band of 1400–1500nm." Usually, positions of peaks of DOS in the energy gap do not directly connected to absorption or luminescence bands in PBE calculations! Authors should correct the corresponding text."
you replied:
"Answer:This error has been modified."
However, after your modification, it is still written (lines 229 - 231): "In the case of defect aggregation, the absorption peak reached 1.13 at 0.75eV, while the absorption peak of POL defect was only 1.08 at 1.07eV,..."
This is incorrect! The positions of peaks in DOS cannot determine the position of absorption or luminescence bands!
So, I recommend authors the following corrections.
Instead of red text in lines 229 -233, it is better to write before line 222: "The calculated DOS presented in Fig.3, suggests that the enhanced optical absorption at 1600-1700 nm in Bi-doped silica based fibers can possibly be determined by the monovalent Bi atom aggregated with ODC(I) and POL intrinsic defects."
Author Response
- Line 23: instead of "silicon glass" you must write "silica glass". You must make this correction throughout the manuscript.
Answer:"Silicon glass" has been replaced with "silica glass"
- Line 26: instead of "...fiber, which is..." you must write "...fibers, which are..."
Answer:These errors have been modified.
- Line 27: instead of "...is..." you must write "...are also..."
Answer:This error have been modified.
- Line 28: instead "...a reinforcer." you must write "...optical amplifiers."
Answer:This error have been modified.
- Lines 28 - 30: instead of "These rare earth-doped fiber lasers do have one drawback, though, 28 which is that they are unable to adequately span the near-infrared spectrum between 1150 29 and 1500 nm" you must write "However, these rare earth-doped fiber lasers have one drawback: they cannot adequately cover the near infrared spectrum between 1150-29 and 1500 nm."
Answer:This error have been modified.
6.Lines 30, 31: instead of "Therefore, the typical rare earth ion doped fiber amplifier and laser 30 hence have trouble meeting the demands of optical fiber communication." you must write "Thus, typical fiber amplifiers and lasers doped with rare earth ions do not meet the requirements of optical fiber communication."
Answer:This error have been modified.
- Lines 35 -38: instead of "The lack of rare earth ion doped fiber amplifiers and lasers is expected to be compensated for by Bi-doped near infrared (NIR) luminescent glass, which will also likely usher in a new generation of ultra wideband optical amplification materials and laser media [9-11]." you must write "The aforementioned disadvantage of rare-earth doped fiber amplifiers and lasers is expected to be compensated by Bi-doped near-infrared (NIR) luminescent glass, which is also likely to usher in a new generation of materials for ultra-wideband optical amplification and laser media [9-11]."
Answer:This error have been modified.
- Line 46: BiO[Error! Reference source not found.] Please, correct this error.
Answer:This error have been modified.
- Lines 46 - 62: There are some errors with numbers of references. The refs. 12, 13, 15, 19, 20, 21, 22, 23 are omitted.
Refs with numbers 12, 13, 15, 19, 20, 21, 22, 23 are missing.
Answer:These errors have been modified.
- Line 72: instead of "quartz" you must write "silica"
Answer:This error have been modified.
- 11. Line 80: instead of "1FS" you must write "1 fs"
Answer:This error have been modified.
- Line 81: Add information about time (or number of steps) of stabilization the system at 300 K; Add the information on how many steps are used to perform the averaging to obtain density of 2.39 g/cm3.
Answer:the number of steps has been added to the article
- Line 85: instead of "quartz" you must write "amorphous silica".
Answer:This error have been modified.
- Line 90: instead of "quartz model" you must write "silica glass model". You must make this correction throughout the manuscript.
Answer:These errors have been modified.
- Line 96: you must change "ev" to "eV"
Answer:This error have been modified.
- Line 103: correct the ref [306]: there is no such big number in the List of references!
Answer:This error have been modified.
- Line 110: Correct English!
Answer:This error have been modified.
- Line 126: "formation energy of three-, four-, five-, and six-membered rings. The results are displayed in Table 1." The English sentence is not full, and some words are missed!
Answer:These errors have been modified.
- Line 128: "Table 1. Distribution table of formation energy of bismuth atom gap doping."
"...bismuth atom gap doping": What does it mean?
Answer:Gap doping refers to doping in the gap. Compared with substitutional doping, it refers to doping in gap of silica glass mode.Relevant content has been added to the article.
- Is Table 1 for Bi-free system or for Bi-doped system?
Answer:Is a Bi- doped system.
- If values in Table 1 correspond to the system with Bi atom, write clearly in the text: where was this bismuth atom initially placed before energy optimization?
Answer:Relevant content has been added to the article.
- To write where Bi atom is in Figure 1!!!
Answer:Relevant content has been added to the article.
- It is still not clear how the values presented in Table 1 were obtained. You must write in the text clearly how values in Table 1 were obtained using equation (1).
Answer:Relevant content has been added to the article.
- Line 157: to write clearly are values presented in Table 2 related to Bi-free samples! What does it mean "...ODC(I) defects and POL defects doped in different ring positions..."? What does it mean here "...doped..."? Is the doping Bi?
Answer:It means adding defects in defect-free amorphous silica fiber. Relevant content has been added to the article.
- You must explain how curves in Fig. 4 were obtained. Were they calculated or experimentally measured?
If they were calculated, you must write how the calculations were made.
Answer:Relevant content has been added to the article.
- Line 230: you must to write how you calculated the position of the absorption peak at 0.75 eV. If it was obtained but a simple difference of peaks in the Density Of States (DOS) in Fig.3, it is incorrect!!! In general, you cannot calculated exact position of the absorption band from peaks positions in DOS! The calculation of the Energy gap (the different between LUMO and HOMO states in defect free silica glass using PBE functional in about 5 eV instead of 9 eV !
Answer:I'm sorry for the misunderstanding.. The absorption peak position is obtained from the dielectric constant diagram in Figure 4, not from the DOS diagram.
- You must to improve English throughout the manuscript!
Answer:Thank you for your suggestion. We will carefully check the English writing method of the article
- In response to reviewer's comment, which I wrote in the previous time:
"Authors wrote "The absorption peaks at 0.75ev are in the near-infrared band of 1400–1500nm." Usually, positions of peaks of DOS in the energy gap do not directly connected to absorption or luminescence bands in PBE calculations! Authors should correct the corresponding text."
you replied:"Answer:This error has been modified."
However, after your modification, it is still written (lines 229 - 231): "In the case of defect aggregation, the absorption peak reached 1.13 at 0.75eV, while the absorption peak of POL defect was only 1.08 at 1.07eV,..."
This is incorrect! The positions of peaks in DOS cannot determine the position of absorption or luminescence bands!
So, I recommend authors the following corrections.
Instead of red text in lines 229 -233, it is better to write before line 222: "The calculated DOS presented in Fig.3, suggests that the enhanced optical absorption at 1600-1700 nm in Bi-doped silica based fibers can possibly be determined by the monovalent Bi atom aggregated with ODC(I) and POL intrinsic defects."
Answer:I'm sorry for the misunderstanding. The absorption peak position is obtained from the dielectric constant diagram in Figure 4, not from the DOS diagram.
